# Visible Fidelity Collector of a Zooplankton Sample from the Near-Bottom of the Deep Sea

Jing Xiao [1], Jiawang Chen [1,2,*], Zhenwei Tian [1], Hai Zhu [1], Chunsheng Wang [3], Junyi Yang [4], Qinghua Sheng [4], Dahai Zhang [1] and Jiasong Fang [5]

1   Ocean College, Zhejiang University, Zhoushan 310027, China; 21834122@zju.edu.cn (J.X.);
    tzw10@zju.edu.cn (Z.T.); zjuzh@icloud.com (H.Z.); zhangdahai@zju.edu.cn (D.Z.)
2   Center for Evolution and Conservation Biology, Southern Marine Science and Engineering Guangdong
    Laboratory (Guangzhou), Guangzhou 511458, China
3   Marine Ecology and Environment Laboratory, Second Institute of Oceanography, MNR,
    Hangzhou 310012, China; wangsio@sio.org.cn
4   College of Electronics and Information, Hangzhou Dianzi University, Hangzhou 310018, China;
    junyiyang@hdu.edu.cn (J.Y.); sheng7@hdu.edu.cn (Q.S.)
5   Shanghai Engineering Research Center of Hadal Science and Technology College of Marine Sciences,
    Shanghai Ocean University, Shanghai 201306, China; jsfang@shou.edu.cn
*   Correspondence: arwang@zju.edu.cn; Tel.: +86-0580-2092214

**Abstract:** The multi-net visible fidelity zooplankton collector is designed to obtain near-bottom fidelity zooplankton. The collector is sent to the designated sampling location based on the information provided by the camera and altimeter. The host computer sends instructions to control the opening of the net port for sample collection and closing of the sampling cylinder cover after sampling. The collector contains three trawls so that three samples can be collected for each test, and environmental parameters can be collected simultaneously. After sampling, The sample maintains its fidelity, that is, maintaining the temperature and pressure of the seabed sample after sampling. Two experiments were carried out in the Western Pacific, and six bottles of zooplankton samples were successfully obtained. The development of a multi-net visible zooplankton collector is of great significance for the collection of near-bottom zooplankton.

**Keywords:** near-bottom zooplankton; multi-net; visible sampling; fidelity; deep sea

## 1. Introduction

Knowledge of the biology and ecology of deep-sea organisms is still very limited compared with all other marine ecosystems [1,2]. Among all deep-sea habitats and domains, knowledge of the planktonic component is far more limited than for the benthic counterpart [3]. Zooplankton biodiversity decreases with increasing water depth, but the equitability increases [4]. The high cost of shipping times and technologies to operate in deep-sea environments makes it difficult to conduct oceanographic sampling [5]. This is particularly evident for investigations on deep-sea zooplankton [6]. Meso- and macro-zooplankton organisms play a key role in biological processes in all marine ecosystems, being a "linkage" between phytoplankton/micro-zooplankton and the higher trophic levels. In addition, zooplankton organisms are able, through vertical migration, to contribute to the functional linkage between the photic zone and the dark deep ocean [7–9].

Sophisticated sampling systems are now available to quantify the abundance of planktonic organisms [10]. The development of electronically controlled multiple net units designed to allow sampling in discrete depth strata has revolutionized our ability to determine the vertical structure and depth-integrated abundance of zooplankton. Less sophisticated net-based sampling devices, however, remain in widespread use, both because of their low cost and their ease of deployment. The well-known ones were developed by the U.S. GLOBEC program (U.S. GLOBEC is a multi-disciplinary research program designed by

oceanographers, fishery scientists, and marine ecologists) with the Bongo and the Multiple Opening and Closing Net Environmental Sensing System (MOCNESS) [11,12] and Bedford Institute of Oceanography Net and Environmental Sampling System (BIONESS) [13]. One sampling type adopts a louvered structure, uses the weight of the net pole to realize the opening and closing of the net port, and uses the diagonal drag method for sampling. The tripping of the net pole is controlled by a device composed of a stepping motor, and the net port is opened when the net pole moves to the bottom of the frame. However, if the sea conditions are not good, the net pole cannot maintain its balance during the fall process and it will get stuck. Another sampling type is Multinet [14], which uses an "inverted L" type opening and closing method.

However, these collectors do not consider the preservation of samples with fidelity, that is, they do not store the collected samples with in-situ insulation and pressure preservation. A sample collected in this way is very different from an in-situ sample. Therefore, research on a visible sampling technique for near-bottom fidelity zooplankton in the deep sea will have important theoretical and practical significance for obtaining live hydrothermal zooplankton.

## 2. Methods

### 2.1. Multi-Net Visible Collector

The specific indicators of the developed multi-net visible zooplankton collector are as follows:

- Maximum design working depth: 4000 m;
- Number of trawl nets: 3;
- Network port area: 0.5 m$^2$
- The effective volume of the sample barrel: $\geq$0.25 L;
- The pressure in the sample storage bin shall not be lower than 80% of the original pressure at the sampling point within 6 h after boarding;
- The temperature rise in the sample storage bin after boarding does not exceed the original temperature of 8 °C;
- Hybrid transmission of the underwater power supply and data images are realized with transmission power $\geq$1.5 kw and transmission distance $\geq$6 km;
- The environmental parameters transmitted in real time include temperature, salinity, depth, turbidity, and dissolved oxygen;
- The total weight of the trailer body: 1.2 tons;
- Outer frame size: 1.5 m × 1 m × 2 m.

### 2.1.1. Sampling Principle

The multi-net visible zooplankton collector consists of two parts: the main body and the sample collection. The overall structure diagram and main body diagram are shown in Figures 1 and 2. After the research vessel reaches the working position, the collector is lowered into the water and sent to the designated sampling location based on the information provided by the camera and altimeter. The motor is sent a command to open a net port through the spring trigger opening and closing mechanism (Figure 3). After the net port is opened, the opening and closing mechanism sends feedback information, the motor stops rotating, and the sampling operation starts. After sampling, the fuse is energized and blown by the command sent, and the sampling cylinder cover is closed under the action of the torsion spring. After the cylinder cover is closed, the opening and closing net mechanism closes the net port under the drive of the motor, and the first sampling ends. After the first sampling is over, the collector can be dragged to the next sampling location, and the above steps are repeated for the second and third sampling in sequence, or the collector is taken out of the water directly.

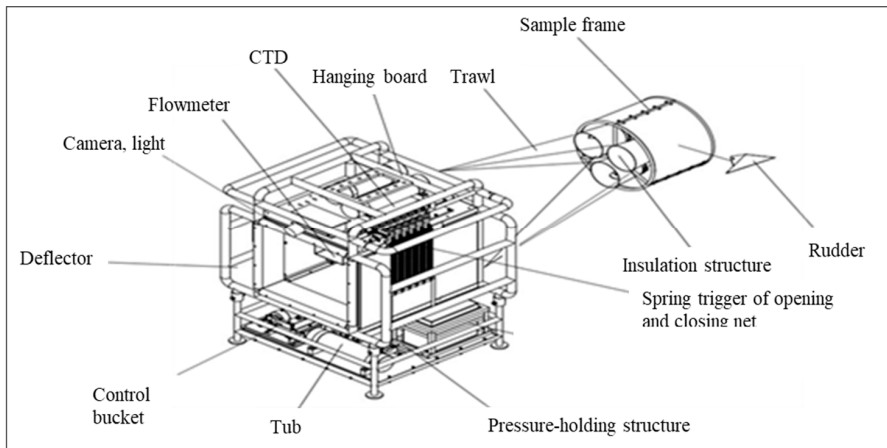

**Figure 1.** The overall structure of the multi-link visual control large-caliber trawl system.

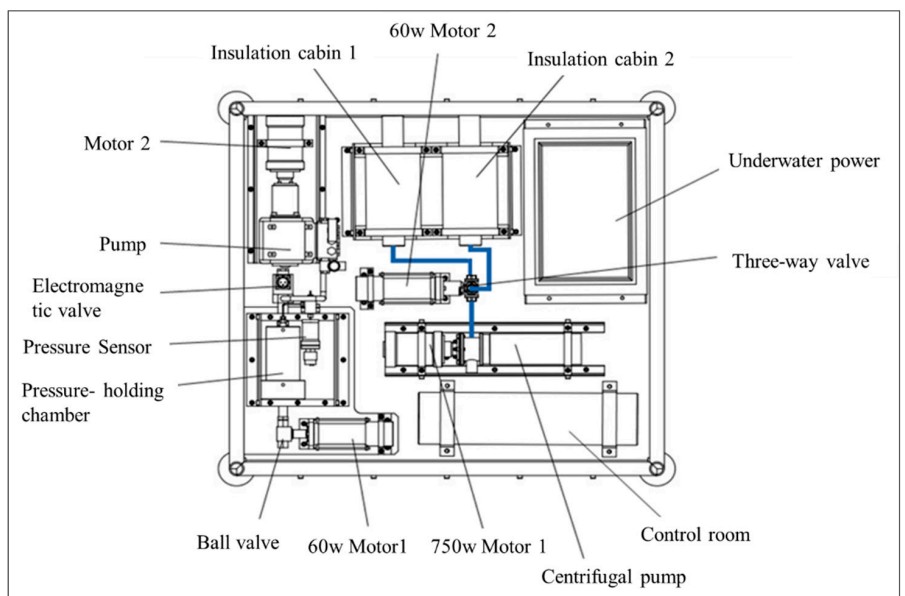

**Figure 2.** The main body of the multi-link visual control large-caliber trawl system.

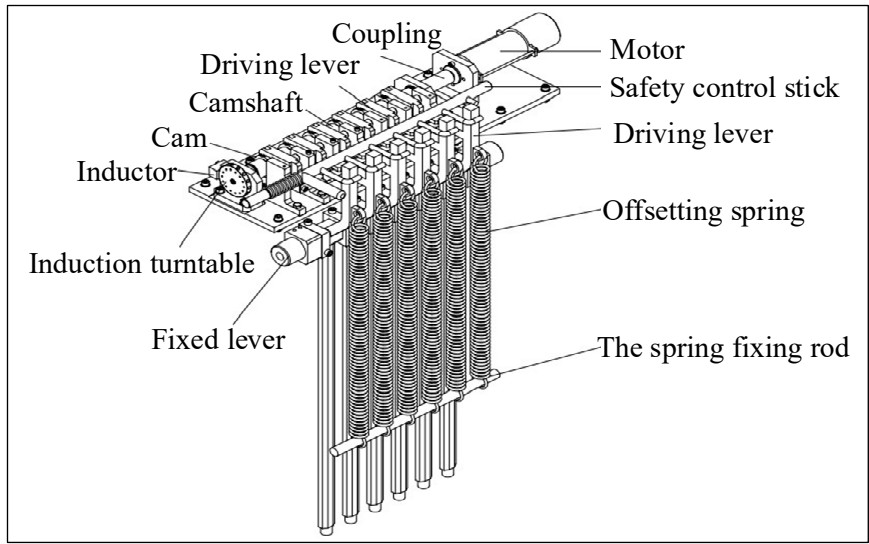

**Figure 3.** Net opening and closing mechanism.

### 2.1.2. The Spring-Triggered Switch

The spring-triggered opening and closing structure (Figure 3) includes a frame, a 60 W brushless DC motor, a camshaft with six cams, three nets, six net port levers, an induction turntable with 14 magnets (magnetic N and S poles are installed crosswise), a sensor, and a locking mechanism with six paddles.

The three net ports are four-sided when they are opened. Two sides are fixed on the trawl support rods, and the other two sides are fixed on two net port levers (one is the opening lever and the other is the closing lever). In the driving mechanism of the opening and closing net (Figure 4), the motor is connected to the camshaft through a coupling, and six cams and an empty station are evenly distributed on the camshaft. The other end of the camshaft is connected to the sensing structure (Figure 5), and the sensor is installed on the sensing plate. The locking mechanism (Figure 6) is fixed on the frame, one end is in contact with the cam, and the other end is locked by the hook. The six levers are fixed on the frame through the middle hole to form a lever mechanism. The other side is connected to the frame by a spring, and the spring is in a stretched state in a non-working state.

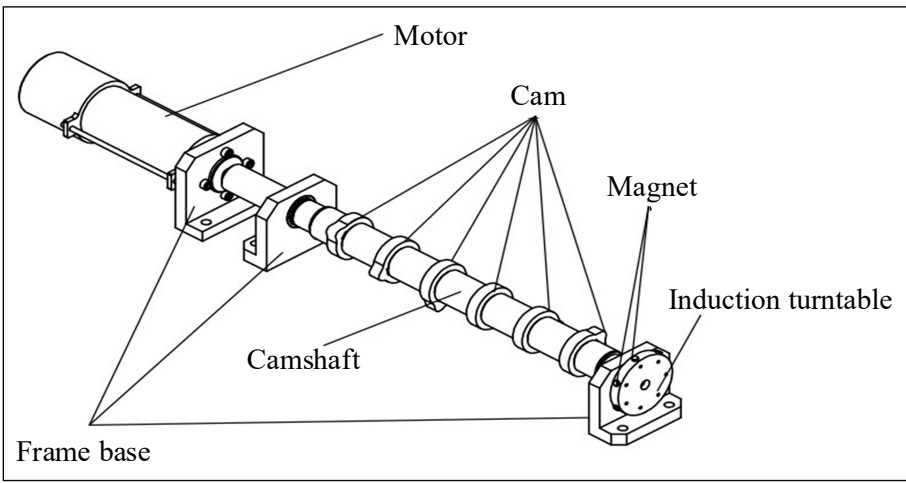

**Figure 4.** Driving schematic diagram of the net opening and closing mechanism.

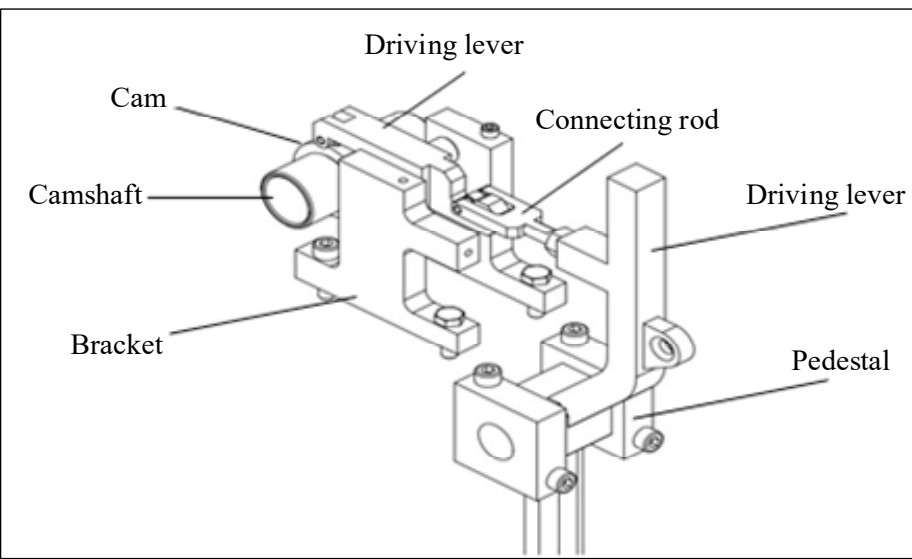

**Figure 5.** Schematic diagram of locking device of opening and closing net mechanism.

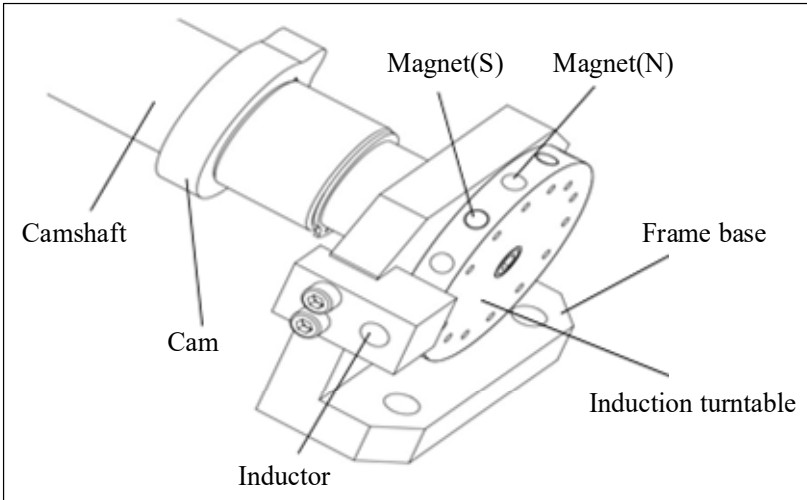

**Figure 6.** Schematic diagram of the induction device of the net opening and closing mechanism.

*2.2. Temperature Retention System*

In order to maintain the original temperature of the zooplankton sample, a sample collection cylinder that can maintain temperature is designed as shown in Figure 7 (the left picture is the front and the right is the back). This part includes an outer frame, three insulation barrels, and a steel wire fixing seat. The sample collector is designed based on the principle of double-layer water bath insulation, and the base material adopts engineering plastics, which has a good insulation effect. The tail sample collection device uses plastic with poor thermal conductivity, uses a solenoid valve to trigger the collection of biological samples, and uses a water bath for sample insulation.

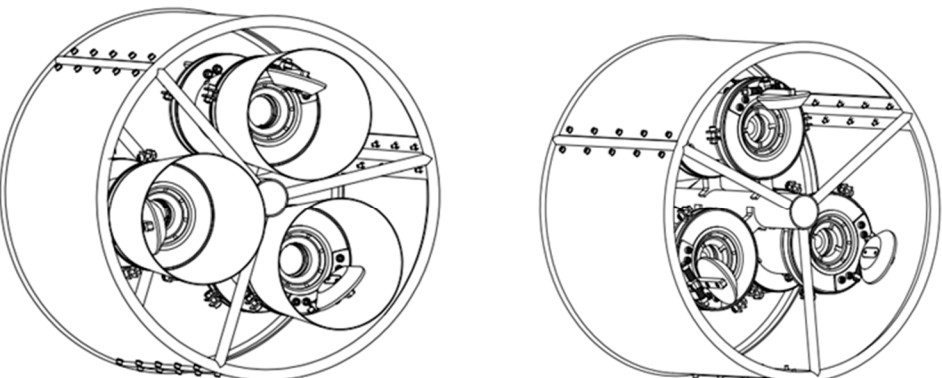

**Figure 7.** Schematic diagram of the sample collection cylinder structure.

The structural cross-sectional view of the temperature-retaining cylinder is shown in Figure 8. It is divided into an outer tube and an inner tube with a filter screen. A cylinder cover is arranged on each end cover, and a torsion spring is installed between each cylinder cover and the end cover. When the cylinder cover is opened, the two covers are connected by steel wire and a fuse, and the torsion spring is in a compressed state. When the fuse is energized and blown, the two cylinder covers are closed under the action of their respective torsion springs, and the sea water and zooplankton samples are sealed in the cylinder.

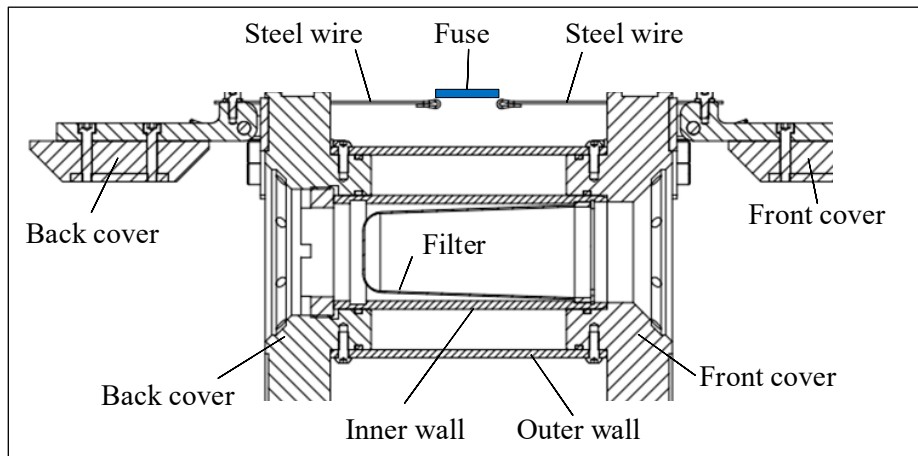

**Figure 8.** Cross-sectional view of the insulation tube.

*2.3. Control System*

The monitoring system of the visible collector mainly includes a deck monitoring unit, an optical cable communication machine, an underwater optical fiber communication module, an embedded control system main board, and driving cabins. The system structure diagram is shown in Figure 9.

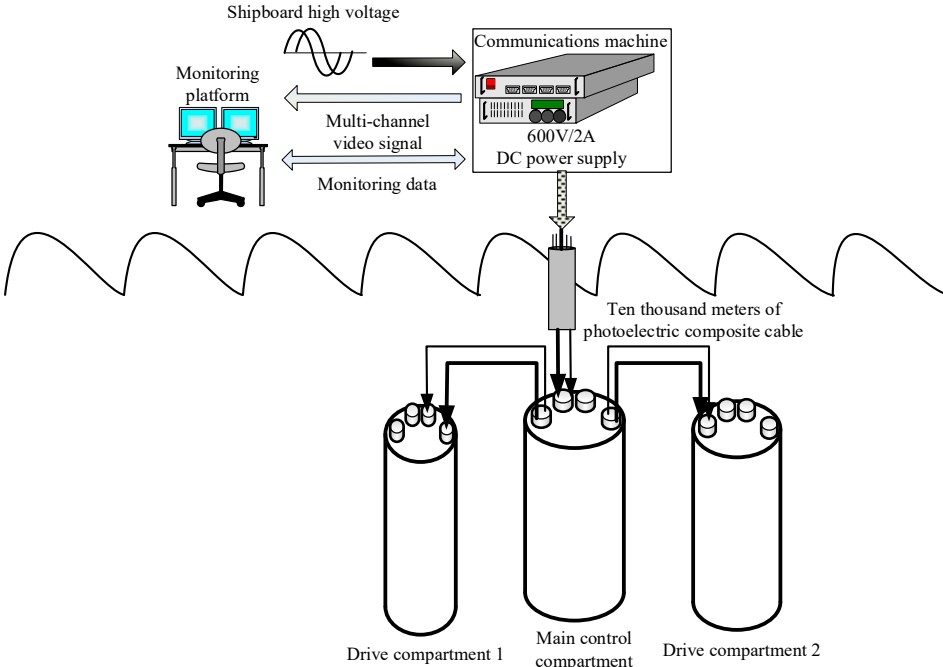

**Figure 9.** Sampling monitoring system structure diagram.

The power supply system converts the shipborne AC high-voltage power into a 600 V/2 A DC stabilized power supply through a DC stabilized power supply and transmits the power to the subsea equipment through an armored photoelectric composite cable. The video image signal of the seabed, the sensor data of the seabed uplink and the feedback data of the action execution result, and the control command data of the upper computer's downlink are transmitted to the seabed through the optical fiber. The monitoring platform realizes the real-time monitoring of the direct-view sampling of near-bottom zooplankton and is responsible for controlling the operation status of the seabed trawl based on seabed monitoring data and video images, as well as drawing the real-time change curve of CTD temperature, conductivity, and pressure (depth) data.

The principle block diagram of the main control cabin is shown in Figure 10, which mainly shows the transmission of direct current, the detection of water leakage, inclination, and flow conditions, the switch control of underwater lights, underwater cameras and bottom altimeters, and the upload of underwater video signals and underwater data.

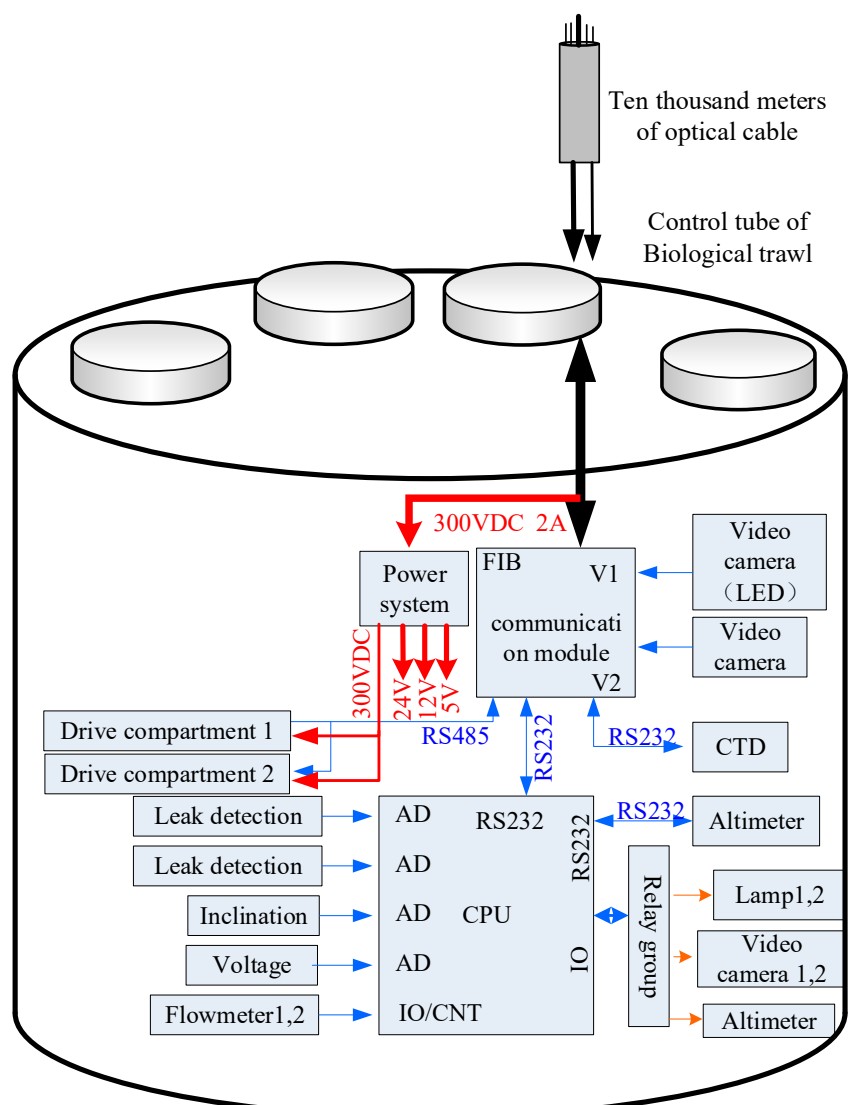

**Figure 10.** Block diagram of the main control cabin.

## 3. Results

### 3.1. The Field Assessment

The multi-net visible zooplankton collector was set up in two sampling stations with water depths exceeding 3000 m in the Northwest Pacific. The sampling device was tested on deck before launching. Through the photoelectric composite cable, the following were carried out: communication test, network port opening and closing test, ball valve opening and closing test, and the altimeter, camera lighting system, flowmeter detection and other functional testing tests. After the research vessel arrived at the predetermined station, its course was adjusted to the top wind and top current, maintaining a speed of 1.5–3 knots with straight sailing. The cable was unwound at a speed of about 30 m/min to a depth of 3000 m, and then the cable was slowly unwound to a distance of 5–10 m from the seabed, after which the cable laying was terminated. The following steps were conducted: Turn on the CTD and other sensors to record and synchronize environmental data during the

unwinding process. Keep the speed and heading stable during the near-bottom operation. Turn on the camera and underwater lights, as well as the video and data recording system.

When the multi-net visible zooplankton collector is tested near the seabed, open the first layer of nets, start trawling for 15 min, close the first layer of nets and open the second layer of nets. Continue to trawl for 15 min, close the second layer of nets and open the third layer of nets. After 15 min of trawling, close the third layer of nets. When the sampling is over, start to recover the collector, raise the cable at a speed of about 30 m/min, and slow down when the collector comes out of the water. After the collector is recovered on deck, check the pressure and temperature in the sample cylinder and determine whether the sample is successfully collected to verify the feasibility of the collector.

MACTD16-1-BPTV station carried out the first operation in the seamount area of the Northwest Pacific, with a maximum working depth of 3487 m and an accumulated underwater working time of about 4 h. MACTD16-BPTVA station carried out the second operation in the seamount area of the Northwest Pacific, with a maximum working depth of 3764 m; the cumulative underwater working time was about 4 h. During the two operations, a total of six samples of near-bottom zooplankton were collected, and synchronized video and environmental data were obtained. The collector was kept in good condition during the operation. The test site is shown in Figure 11.

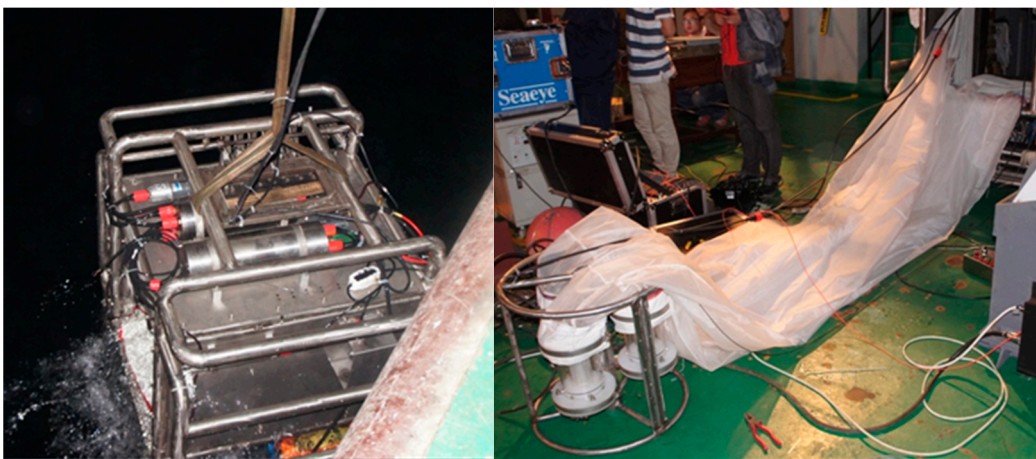

**Figure 11.** The sea trial site of the collector.

*3.2. Experimental Results*

The multi-net visible zooplankton collector was used to conduct two sets of experiments at two stations. After the sampling was completed, the data obtained through CTD included the water depth, pressure, and temperature of each station at the time of sampling. The records are shown in Table 1.

**Table 1.** Data record table.

| Sample No. | Sampling Time | Working Depth | Pressure (MPa) | Temperature (°C) | Pressure (MPa) (6 h Later) | Temperature (°C) (6 h Later) | Sampling |
|:---:|:---:|:---:|:---:|:---:|:---:|:---:|:---:|
| 1 | 15 min | 2685 | 26.00 | 3.369 | 24.2 | 10.10 | Yes |
| 2 | 15 min | 3487 | 34.00 | 3.020 | 31.0 | 7.60 | Yes |
| 3 | 15 min | 3016 | 0.35 | 17.200 | / | / | Yes |
| 1 | 15 min | 3205 | 32.00 | 3.112 | 29.8 | 8.10 | Yes |
| 2 | 15 min | 3764 | 37.00 | 3.031 | 34.2 | 7.78 | Yes |
| 3 | 15 min | 3341 | 33.00 | 3.117 | 29.5 | 7.82 | Yes |

It can be seen from the data in Table 1 that in the first sampling experiment, the internal pressure was not maintained due to the closure failure of the No. 3 network port, and the temperature was also the sea surface temperature at that time. After inspection and adjustment, the collector was lowered again. In the second sampling, the network ports of the three sample cylinders closed normally. The maximum working depth of the collector was about 3487 m. The sensors in the chambers showed that the pressure-retaining effect was good, and the pressure and temperature data were also obtained successfully. Six hours after the samples were taken, the pressure and temperature data of the sample cylinders were used to verify the performance of the device and the sample acquisition capabilities.

According to the pressure gauge, the pressure of the sample cylinders after 6 h was higher than 80% of the initial pressure. The water temperature on the seafloor was about 3 °C, and the maximum temperature in the sample cylinder after the collector was recovered on deck was 10.1 °C, an increase of 7.1 °C. The pressure and temperature data showed that the samples were maintained at their original pressure and temperature.

The fidelity samples were sent to a biological laboratory for research; the collected samples could not be viewed directly because opening the lid will cause the sample to lose its original temperature and pressure. Through the samples remaining in the trawl net and in the sample cylinder No. 3 in the first test (Figure 12), it was determined that the sample was obtained successfully.

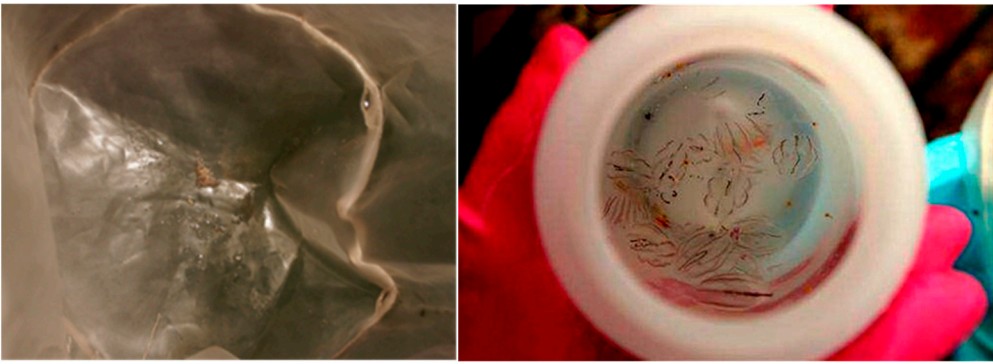

**Figure 12.** Samples in the trawl net and sample tube #3.

## 4. Discussion and Conclusions

### 4.1. Analysis of Failure to Close the Trawl Net Port

For this substantial sampling test, the sampling success rate of the two experiments was 100%, and the success rate of retaining the sample in the in-situ environment was 83.3%. The main reason that led to the failure of the No. 3 network in the first test was the failure of the No. 3 lever spring when it was reset. Every two levers control the opening and closing of a trawl, and each lever uses the contraction force of the spring to complete a 90° rotation to complete the opening and closing of the net port. As shown in Figure 13a, in the initial state, both pole 1 and pole 2 remain vertical, and the four nodes of the network port are fixed at four respective points: A, B, C, and D (the thick black line in Figure 13 represents the network port), where point A is fixed on the bottom plate, point B is on rod 1, and points C and D are on the upper and lower ends of rod 2. When the motor sends a rotation signal, the camshaft rotates by an angle to disengage the claw sleeve and the claw of pole 1. Rod 1 rotates 90° around the axis to reach the position shown in Figure 13b under the action of the spring force, so that net 1 is fully opened into a large "mouth" shape and enters the working state. When the motor gives a rotation signal again, the camshaft rotates by an angle again, which triggers pole 2 to rotate 90°, reaching the position shown in Figure 13c, and completing the closing of trawl 3. However, due to the failure of the spring reset, network port 3 is always kept open, so the temperature and pressure of the sample cannot be maintained.

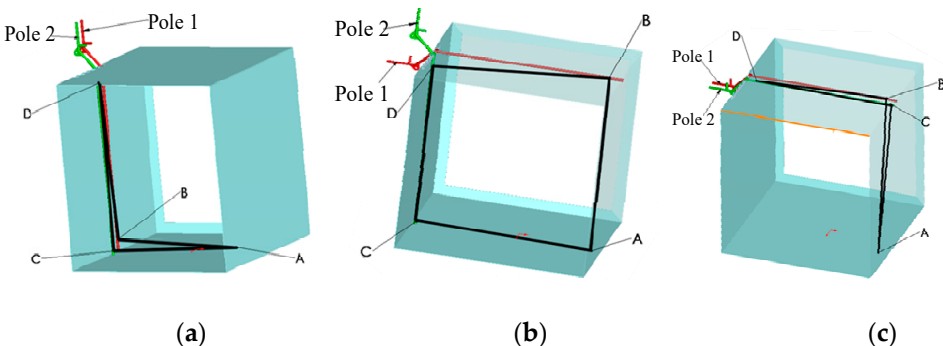

**Figure 13.** Samples in the trawl net and sample tube #3. (**a**) Initial state, (**b**) Working state, (**c**) Final state.

Since the work of the three trawls is relatively independent, their work sequence is determined by the movement of the levers. The bounce sequence and time interval of the levers can be freely controlled by arranging the position of the cam so that the opening and closing of the trawls can be controlled.

### 4.2. Conclusions

The multi-net visible zooplankton collector for near-bottom of the deep sea can carry out large-volume drag sampling and can maintain the pressure and temperature of the original environment of the sample. During the sampling process, the operation can be monitored by the camera, and environmental parameters can be collected simultaneously. In the Northwest Pacific experiment, five bottles of zooplankton samples were obtained, and the temperature and pressure of the samples were retained. The multi-net visible zooplankton collector provides a good start for obtaining near-bottom fidelity zooplankton and will be used for fidelity research of deeper-sea organisms.

**Author Contributions:** Conceptualization, J.X., J.C. and Z.T.; methodology, J.C.; software, J.X.; validation, Z.T., H.Z.; formal analysis, J.X., J.C.; investigation, H.Z.; resources, Z.T.; data curation, J.X.; writing—original draft preparation, J.X.; writing—review and editing, J.X., C.W.; visualization, J.Y.; supervision, Q.S.; project administration, D.Z.; funding acquisition, J.F. All authors have read and agreed to the published version of the manuscript.

**Funding:** The research was funded by the National Key R&D Program of China (grant number 2018YFC0310600) and Key Special Project for Introduced Talents Team of Southern Marine Science and Engineering Guangdong Laboratory (Guangzhou), China (GML2019ZD0506).

**Institutional Review Board Statement:** Not applicable.

**Informed Consent Statement:** Not applicable.

**Data Availability Statement:** Not applicable.

**Acknowledgments:** The study is based on the National Key R&D Program of China (2018YFC0310600) supported by the Ministry of Science and Technology of the People's Republic of China and the Key Special Project for Introduced Talents Team of Southern Marine Science and Engineering Guangdong Laboratory (Guangzhou), China (GML2019ZD0506).

**Conflicts of Interest:** We declare that we have no financial and personal relationships with other people or organizations that can inappropriately influence our work, and there is no professional or other personal interest of any nature or kind in any product, service and company. The manuscript entitled, "Visible fidelity collector of zooplankton sample from the near-bottom of the deep sea".

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
