# Peer review of "Visible Fidelity Collector of a Zooplankton Sample from the Near-Bottom of the Deep Sea"

_jmse, doi:10.3390/jmse9030332_

Round 1

Reviewer 1 Report

It is clear that a lot of effort has gone into designing, building and testing this plankton sampling device. However, it is not clear why. The requirements for the device are not presented and the benefits are not clear. The arguments made for retaining larvae at pressure are not coherent or well structured. The tie in with deep-sea hydrothermal vent survey is weak with arguments that are not substantiated with any appropriate references to existing studies. Using towed plankton nets for sampling close to the seabed for vent plume larva is not a realistic prospect, vent plumes are warm so tend to be positively buoyant. It is very close to impossible to identify a vent larva without genetics, even with a microscope it is hard due to a lack of keys. Furthermore, by placing the sample under the microscope, the pressure effect is lost, so you might as well have used a traditional sampling device. Sampling plankton closer to a vent is best carried out by an ROV.

There are serious errors with referencing throughout the document. I highly recommend that the lead authors read some literature relevant to the subject matter prior to submission for review in an international peer-reviewed journal.

The authors appear to be trying to push this paper towards relevance for deep-sea hydrothermal vent research without any evidence. The authors demonstrate little competence or understanding of deep-sea sampling with an almost total lack of knowledge of the literature (only a single appropriate reference). 

There is no evidence presented here that nether is anything novel about this system or that it is in any way more efficient or useful than existing techniques. They have not presented evidence that the system actually works  - there is no evidence that the camera system added anything or that the triggering mechanism is more reliable in inclement weather. They appear to have re-engineered a MOCNESS with a slightly different triggering and net opening mechanism. 

I suggest a major rewrite.

Line 25-26 [1]: This is not an appropriate reference to support this rather vague statement – the paper makes no reference to hydrothermal vents.

Line 27 [2]: This is not an appropriate reference to support this statement – the paper makes no reference to hydrothermal vent communities. Also, the statement is lacking focus, issue, specifically pertaining to this paper need to be addressed? 

Line 30 [3,4]: These are not appropriate references to support this statement – nether paper discusses hydrothermal vents.

Line 32 [5]: This is not an appropriate reference to support this statement – for the record Pomoxis are small Northern American freshwater fish not hydrothermal endemic species – they are known as ‘Sunfish’ (the name might be a good hint that they are not deep-sea hydrothermal vent endemic fauna). Furthermore, cultivating hydrothermal vent larva does clarify much and is incredibly difficult. A lot can be garnered from preserved samples - the larva can be identified (genetics) and a lot of their life history can be inferred from their physiology (lecithotrophic, planktotrophic etc)

Line 34 [6,7]: These are not appropriate references to support this statement – nether paper discusses hydrothermal vents.

Line 35 [8]: These are not appropriate references to support this statement – nether paper discusses hydrothermal vents. You have not demonstrated any understanding of vent larval dynamics.

Line36 [9]: This statement is both unsubstantiated (the citation refers to a device for sampling vent fluids and has nothing to do with vent larva) and just incorrect.

Line 38 [10]: the reference is not appropriate, this citation refers to MicroNESS not MOCNESS

Line 39: A CORRECT REFEFRENCE!!! The 11th reference in this paper is appropriate! That I have to make a statement highlighting a single correct reference in this paper says it all.

Line 40: It is unclear what system is you are discussing, MOCNESS or BIONESS?

Line 46: The BIONESS system has a camera.

Line47: The MOCNESS net is rated to 6000m.

Line49-52: What is the relevance of this paragraph to the narrative? You are introducing a paper dealing with a deep-sea plankton sampling device. The devices referenced here are small hand-held devices for shallow water.

Line 53: what is a fidelity device?

Line55 [16]: this is quite an odd reference and not relevant to the paper. It was written in 1965, hydrothermal vents were first discovered in 1977!

Lines56-58: This sentence makes no sense, why would visible sampling allow for any understanding of ‘fidelity’. I am not sure what you mean by fidelity in this context. Identifying the origin of larvae is almost impossible, even stating if a larva is from a vent endemic species would be almost impossible without molecular approaches.

Line 59: This table is full of errors, I highly recommend the authors do some reading around the subject matter.

Line 63: The figure does not differentiate between the ‘main body and the sample collection’. It is not clear what the authors are referring to.

Lines 83: How are the mechanisms, electronics protected from pressure, what materials were used (stainless steel, titanium?).

Line 107: What materials were used how are they sealed, what are the pressure ratings? How is temperature and pressure monitored? Photographs would help.

Line 217 -219: This does not make sense, you have a success rate of 100% but only retained the in-situ environment 83% (on the deployments). The principle purpose of this device (I think) is to return samples at pressure. This does not appear to have worked. Basing any conclusions not two deployments is not satisfactory.

Author Response

Thank you very much for your pertinent suggestions. Your suggestions are of great help in improving the quality of the article. It is also a place worth learning and paying attention to in our writing. The article has been greatly revised, hope to get your approval.

The introduction has been rewritten, and the requirements and benefits of the device have been added in Chapter 2. The original intention of the device design is to be suitable for submarine hydrothermal organisms, but in practice the device is not limited to sampling near the hydrothermal vents. Since the authors are all designer, we don’t know much about these, and we mistakenly adopted the background introduction part of the original project book, we apologize for that. For this revision, we spent a long time rechecking the literature and made all revisions.

In addition, the design of the device is very meaningful. The success mentioned in the article means that the temperature does not exceed the in-situ temperature of the seabed by 8°C, and the pressure is not lower than 80% of the in-situ pressure. Although there is no 100% preservation of the pressure and temperature to the bottom of the sea, there have been great breakthroughs, which provide a good reference for fidelity sampling. As for what you said about releasing pressure during laboratory research, this is a reality, but the device for pressure-holding culture is also being studied. The entire system of fidelity sampling and fidelity culture will become a reality.

The meaning of fidelity is to try to keep the sample in its original temperature and pressure environment, which has been added in the article. Since the devices used in the marine environment must be pressure-resistant and corrosion-resistant, and be as light as possible, we choose titanium alloy materials. In order to achieve heat preservation and pressure preservation, we have designed a suitable structure and wall thickness, and selected a double-sealed form.

We tried our best to improve the manuscript and made some changes to the manuscript. These changes will not influence the content and framework of the paper. We appreciate for your warm work earnestly, and hope that the correction will meet with approval. Looking forward to hearing from you. Thank you and best regards.

Once again, thank you very much for your comments and suggestions.

Reviewer 2 Report

Comment on paper “Visible collection of fidelity zooplankton sample from the near-bottom of the deep sea” by Jing Xiao et al.

The authors present a description and principle of operation of the multi-net zooplankton sampler - a new device for accurate collection of near-bottom plankton. Compared to existing samplers (MOCNESS, BIONESS, MULTI-NET) this instrument has several important advantages, the most important of which is the ability to keep the sample at ambient temperature and pressure. In addition, this sampler is equipped with an altimeter and a video camera, allowing for more accurate collection of zooplankton from the near bottom depth.

As a biologist, I cannot confidently evaluate this device from a technical point of view, but I can say that the biological material collected by such a sampler can be of great importance for laboratory experiments on physiology of deep-water organisms.

Remarks.

  • The term “near-bottom zooplankton” should be used throughout the text instead of “zooplankton larvae” (line 30), “hydrothermal larvae” (line32), “seabed organisms” (line 139), “biological larvae” (line 180), “samples of seabed” (line 250), The use of the listed terms is incorrect and has no biological meaning.
  • Some technical data are missed:
  • dimensions of control sampler system
  • length of net bags
  • mesh size of the netting
  • diameter of net buckets
  • overall length ready for operation (from bridle to net bucket)
  • weight of sampler.
  • Table 1. As far as I know, all MOCNESS systems are capable of sampling to 6000 meters depth.
  • Tables 2 and 3 contain overlapping data.They can be combined into one table.

Number of decimal places must be equal for identical parameters.

Table 3. Why did the pressure in sample No 6 after six hours increase by 6 MPa, while in the rest of the samples it decreased by 2-3 MPa? At the same time, the authors write that “the pressure of the sample cylinders after 6 hours is higher than 80% of the initial” (line 202-203) while it decreased by approximately 8% (excluding sample No 6). After six hours onboard the temperature in the sample cylinders increased from cca 3 ℃ to 7.6-10.1 ℃. However, the authors argue that “The pressure and temperature data show that the samples are maintained their original pressure and temperature”. (line 205-206).

My main concern is the lack of data on the collected biological material that could confirm the effectiveness of the new sampler. The photographs shown in Figure 12 can only raise several questions. Why did the plankton stay in the trawl and not concentrate in the cod end? Why were there only a dozen comb jellies in the cylinder? These photographs only testify to the fact that the net was catching something, as any other net towing from the bottom to the surface would do. In other words, these photographs cannot in any way confirm the authors' conclusion: “Through the samples remaining on the trawl and in the sample cylinder No. 3 in the first test, it can be determined that the sample has been obtained successfully” (lines 209-211).

Minor remarks.

  • Two sentences contradict one another. “Take out the sample and transfer it to the sample bottle, fix it with the fixative, and fill in the sampling record form" (lines 173-174) and “The fidelity samples are sent to the biological laboratory for research, the collected samples cannot be viewed directly, because opening the lid will cause the sample to lose its original survival temperature and pressure” (lines 207-209).
  • It is better to replace “probe ship” (line 65) and “scientific research ship”(line 158) with a common “Research vessel”.
  • The sentence on lines 74-75 repeats the previous sentence on lines 72-73.

Author Response

Thank you very much for your pertinent suggestions and encouragement. Your comments are of great help in improving the quality of the article.

    Regarding the data of sample 6, I must explain to you. Due to my carelessness, I typed the number 2 on the data recording paper into 3 and wrote it in the table. In fact, the data is 29.5. I'm really sorry for making this mistake.

The success mentioned in the article is based on the requirements of the device on the project, that is, after the sample is taken, the temperature does not exceed the in-situ temperature of the seabed by 8℃, and the pressure is not less than 80% of the in-situ pressure. The basic parameters and requirements are written in the second part.

Since the samples were handed over to biologists after collection, the only photos we took were the remaining organisms on the trawl and a sample barrel that was not successfully sealed. The sample barrel is pressure-retained and sealed after sampling on the seabed, and it is impossible to observe whether the sampling is successful, but if there are samples in the field involved, the sample must be successfully obtained by filtering the seawater, although the best way is to tell us that the sampling is successful through the test data of biologists..

We have reorganized other questions about the vocabulary, sentences and tables and tried our best to improve the manuscript and made some changes to the manuscript. These changes will not influence the content and framework of the paper. We appreciate for your warm work earnestly and hope that the correction will meet with approval. Looking forward to hearing from you. Thank you and best regards.

Once again, thank you very much for your comments and suggestions.